computational mechanics/computer modelling and simulation

ascending aorta aneurysm, Marfan syndrome, computational fluid dynamic, fluid–structure interaction, shear stress ratio

**Author for correspondence:**
J. Martorell
e-mail: jordi.martorell@iqs.edu

[†]Present address: Vascular Engineering and Applied Biomedicine, IQS School of Engineering, Universitat Ramon Llull, Via Augusta 390, 08017 Barcelona, Spain.

# Fluid–structure interaction simulations outperform computational fluid dynamics in the description of thoracic aorta haemodynamics and in the differentiation of progressive dilation in Marfan syndrome patients

R. Pons[1], A. Guala[2], J. F. Rodríguez-Palomares[2], J. C. Cajas[3,4], L. Dux-Santoy[2], G. Teixidó-Tura[2], J. J. Molins[1], M. Vázquez[3,5], A. Evangelista[2] and J. Martorell[1,†]

[1]Department of Chemical Engineering and Material Sciences, IQS School of Engineering, Universitat Ramon Llull, Via Augusta 390, 08017 Barcelona, Spain
[2]Hospital Universitari Vall d'Hebron, Department of Cardiology, CIBER-CV, Vall d'Hebron Institut de recerca (VHIR), Universitat Autonoma de Barcelona, Barcelona, Spain
[3]Barcelona Supercomputing Center (BSC-CNS), Department of Computer Applications in Science and Engineering, C/Jordi Girona 29, 08034 Barcelona, Spain
[4]Escuela Nacional de Estudios Superiors, Unidad Mérida, Universidad Nacional Autónoma de México, Carretera Mérida-Tetiz, Km 4, Ucú, Yucatán, 97357, México
[5]ELEM Biotech, Calle Rossello 36, 08029 Barcelona, Spain

RP, 0000-0003-0040-928X; LD-S, 0000-0001-8736-2095; MV, 0000-0002-2526-6708; JM, 0000-0002-2043-2762

Abnormal fluid dynamics at the ascending aorta may be at the origin of aortic aneurysms. This study was aimed at comparing the performance of computational fluid dynamics (CFD) and fluid–structure interaction (FSI) simulations against four-dimensional (4D) flow magnetic resonance imaging (MRI) data; and to assess the capacity of advanced fluid dynamics markers to stratify aneurysm progression risk. Eight Marfan syndrome (MFS) patients, four with stable and four with dilating aneurysms of the proximal aorta, and four healthy controls were studied. FSI and CFD simulations were

performed with MRI-derived geometry, inlet velocity field and Young's modulus. Flow displacement, jet angle and maximum velocity evaluated from FSI and CFD simulations were compared to 4D flow MRI data. A dimensionless parameter, the shear stress ratio (SSR), was evaluated from FSI and CFD simulations and assessed as potential correlate of aneurysm progression. FSI simulations successfully matched MRI data regarding descending to ascending aorta flow rates ($R^2 = 0.92$) and pulse wave velocity ($R^2 = 0.99$). Compared to CFD, FSI simulations showed significantly lower percentage errors in ascending and descending aorta in flow displacement (−46% ascending, −41% descending), jet angle (−28% ascending, −50% descending) and maximum velocity (−37% ascending, −34% descending) with respect to 4D flow MRI. FSI- but not CFD-derived SSR differentiated between stable and dilating MFS patients. Fluid dynamic simulations of the thoracic aorta require fluid–solid interaction to properly reproduce complex haemodynamics. FSI- but not CFD-derived SSR could help stratifying MFS patients.

## 1. Introduction

Thoracic aortic aneurysms are normally associated with conditions such as hypertension, ageing or genetic abnormalities, like Marfan syndrome (MFS) [1]. Most cases are asymptomatic and only diagnosed as incidental findings. Spontaneous rupture is often fatal and hence patient management after diagnosis is critical [2]. According to clinical guidelines, the maximum aortic diameter is the main parameter used for the assessment of rupture risk. Surgical intervention is indicated for ascending aorta aneurysms with diameters larger than 50 mm in MFS patients [3,4] or 55 mm in non-genetic aortic aneurysms. However, considering only maximum aortic diameter has demonstrated to be ineffective. Indeed, results from a large international registry of acute aortic dissection (IRAD) showed that around 40% of life-threatening events like aortic dissection happened with aortic diameters below those recommended for surgical intervention [5]. Remarkably, multimodality imaging has agreement and reproducibility limitations [6]. In this context, the use of biomechanical markers, such as aortic wall shear stress (WSS), stiffness and strain, is gaining a prominent role in the quest of possible factors for improving patient stratification [7–11]. Recent studies have shown that the region of high WSS matched those with high extracellular matrix dysregulation and fibre degeneration [12], thus providing evidence that role flow abnormalities may contribute to aneurysm progression [9,11,13]. However, to date, biomechanical markers have not been included in clinical practice.

Mathematical and computational models can provide several biomechanical factors in aortic aneurysms. Clinical imaging techniques like computed tomography and magnetic resonance imaging (MRI) are key to build reliable computational three-dimensional (3D) aortic models [14], which are used to compute patient-specific WSS and haemodynamics. Computational fluid dynamics (CFD) analysis of aortic flow not only needs a tri-dimensional detailed vessel geometry, but also inlet and outlet velocity profiles [15], which is to date a major issue to obtain accurate fluid flow predictions [16]. Four-dimensional (4D) phase-contrast cardiovascular magnetic resonance (CMR) (4D flow MRI) imaging has made it possible to non-invasively quantify patient-specific *in vivo* blood velocity profiles. However, there are serious limitations in both spatial and temporal resolution of the signals and data are corrupted by noise-like phase error [17]. Recently, multiple studies have used 4D flow MRI for the assessment of haemodynamics of the aorta evaluating aortic flow patterns, WSS and regional aortic stiffness (via pulse wave velocity (PWV) in the thoracic aorta of bicuspid aortic valve [8,18–20] and MFS [21,22] patients). Outlet boundary conditions can be simulated using the Windkessel model. This model describes the behaviour of the whole arterial system distal to the boundary in terms of a pressure–flow relationship using an electrical analogy for fluid flow [23].

The reliability of CFD for estimation of flow through aneurysms was validated using particle image velocimetry measurements in idealized and realistic models [24]. However, CFD disregards wall motion, e.g. the interaction between pulsatile blood flow and the compliant arterial wall, which may affect the estimation of WSS distribution at the aortic wall, as it does not consider blood flow accumulation during systole and its release in diastole [25,26]. The computation of fluid–structure interaction (FSI) problems has gained relevance in the past decade as large computing platforms have become more available and parallel computing has significantly evolved. In FSI models, the interaction of a deformable structure (here the vessel walls) with an internal fluid flow (the blood flow) is computed. Aortic stiffness, a fundamental parameter in FSI models, can be retrieved from PWV through the Moens–Korteweg equation [27].

Few FSI studies have been performed on anatomically realistic aneurysms models to date [28–30]. Torii *et al.* investigated FSI in two cerebral aneurysms showing that wall deformation affects the distribution of WSS. Also, it has been shown that CFD simulation underestimated WSS on average by 10–30% compared to FSI [14]. However, to our knowledge to date, no FSI studies have been published in MFS patients, which are known to have abnormal aortic wall mechanical properties [8] affecting aortic wall deformation [10] and aneurysm progression.

## 1.1. Hypothesis and objectives

As the thoracic aorta is subject to remarkable pulsatile displacement, we hypothesized that FSI simulations including MRI data would outperform CFD in the prediction of flow patterns in the thoracic aorta. With accurate metrics, a parameter combining the effect of fluid dynamics and wall mechanics may better characterize the prognosis of ascending aorta aneurysms. Therefore, the main objectives of this work are (i) to study the accuracy of CFD and FSI simulations compared to 4D flow MRI data in MFS patients with a thoracic aorta aneurysm, and (ii) to compare advanced fluid dynamics markers between derived by CFD versus FSI simulations and assess their capacity to stratify the risk of aneurysm progression.

# 2. Methods

Eight MFS patients with no history of aortic dissection or surgery and free from aortic valve diseases and four healthy controls were retrospectively selected at a tertiary reference hospital. MFS patients were classified according to the past aortic dilation rate as following: four stable (if the growing rate was less than 0.6 mm year$^{-1}$) and four dilating (if it was greater than or equal to 0.6 mm year$^{-1}$). This threshold was established based on the reported average growth rate of MFS patients [31].

A radially undersampled acquisition (phase contrast–isotropic voxel radial projection imaging) with five-point balanced velocity encoding [32] was used for 4D flow imaging of the entire thoracic aorta without intravenous contrast agent. Data were acquired using the following parameters: velocity encoding 200 cm s$^{-1}$, field of view $400 \times 400 \times 400$ mm, acquisition matrix $160 \times 160 \times 160$, voxel size $2.5 \times 2.5 \times 2.5$ mm, flip angle 8°, repetition time 4.2–6.4 ms and echo time 1.9–3.7 ms. This dataset was reconstructed according to the nominal temporal resolution of each patient (approx. 25 ms). Reconstructions were performed offline with corrections for background phase from concomitant gradients, eddy currents and trajectory errors of the 3D radial acquired *k*-space [32]. A stack of two-dimensional (2D) cine MRI images on double-oblique sagittal plane were collected (figure 1*a*). Brachial systolic and diastolic pressures were taken immediately after the CMR study. Proximal aorta PWV was computed from 4D flow data as previously described [8]. The three cusp-to-commissure diameters were measured by double-oblique cine CMR at the level of the aortic root at end diastole and the largest was considered for the analysis. Ascending aorta diameter was measured at end diastole by double-oblique cine CMR at the level of the pulmonary bifurcation. This study was approved by the internal review boards.

The thoracic aorta, between the sinotubular junction and the descending aorta at the same height, was statically segmented from peak systolic cine MRI image datasets using a proprietary and personalized semiautomatic code in SOLIDWORKS (Dassault Systèmes, France). This code allows lumen and arterial wall reconstruction along the aorta using MRI scan data. Central position, diameter and wall thickness of multiple 2D sagittal slices along the vessels were measured for the reconstruction. The descending aorta and the three supraaortic arteries were artificially extended by 3 cm in the longitudinal direction to ensure flow stabilization at the outlet surface and numerical convergence.

Segmented geometries from SOLIDWORKS were exported as STEP files and imported in GID (Compass Ingeniería y Sistemas, Spain). GID was used to create the finite-element mesh and to apply the corresponding boundary conditions for both domains, the lumen and the wall vessel. The lumen and the wall domains were meshed using unstructured linear tetrahedral elements with a spatial grid of 0.8 mm. An example of a geometry of lumen and wall artery is shown in figure 1. The number of elements for the fluid and the solid is described for all cases in table 1.

Fluid simulations in the thoracic aorta were performed using ALYA, a high performance computational mechanics code able to solve complex coupled multi-physics problems like incompressible/compressible flow, nonlinear solid mechanics, turbulence modelling and biomechanics [33–37]. The multi-code coupled feature of ALYA was used in the present work. The first code solves

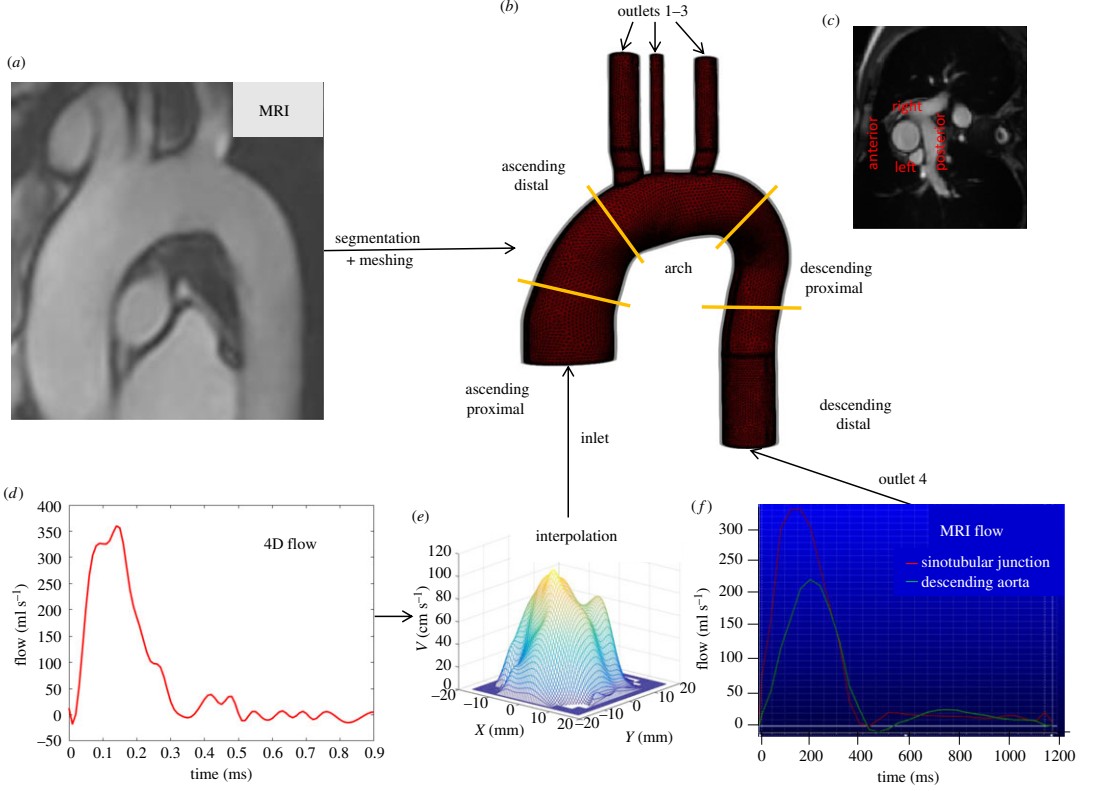

**Figure 1.** Protocol to extract geometric and velocity data. (*a*) MRI is used to extract and segment the geometric characteristics of the aorta. The model is divided into five segments (*b*) and the ascending aorta is divided into four quarters (*c*). Sinotubular junction flow extracted from 4D flow MRI (*d*) and interpolated spatially and temporally (*e*) to use as inlet boundary condition for the simulation. Total flow at the sinotubular junction and the descending aorta are read using MRI (*f*). Solid, fluid and boundary conditions are meshed using GiD.

**Table 1.** Number of fluid and solid elements for each studied case. (C1–4 are controls, S1–4 are stable patients and D1–4 are dilating patients.)

| | | mesh | |
|---|---|---|---|
| | case | fluid elements | solid elements |
| controls | C1 | 1 937 562 | 437 982 |
| | C2 | 1 157 132 | 450 512 |
| | C3 | 1 118 413 | 446 210 |
| | C4 | 1 015 969 | 416 769 |
| stable | S1 | 1 159 257 | 433 867 |
| | S2 | 894 270 | 468 258 |
| | S3 | 1 040 036 | 418 282 |
| | S4 | 887 049 | 447 913 |
| dilating | D1 | 1 330 184 | 612 437 |
| | D2 | 1 914 810 | 659 591 |
| | D3 | 1 397 577 | 495 744 |
| | D4 | 1 912 479 | 565 772 |

the fluid mechanics problem, while the second one solves the solids mechanics case. The codes are coupled using parallel techniques [38]. All simulations were performed using the Mare Nostrum IV supercomputer; 72 processor cores, 48 for the fluid and 24 for the solid were used for FSI simulations.

For fluid dynamics calculations [39], the Navier–Stokes equations (2.1) were discretized using the stabilized finite-element method with variational multi-scale stabilization. The momentum equation is separated from the continuity equation (2.2) using the Schur complement for pressure, each equation is solved independently, and the solution of the coupled system is obtained iteratively. The displacement of the fluid domain is achieved using the arbitrary Lagrangian Eulerian formulation. The time integration scheme used was a backward differentiation formula of second-order Navier–Stokes and continuity equations read

$$\rho_f \frac{\partial u_f}{\partial t} + \rho_f[(u_f - u_m) \cdot \nabla]u_f - \nabla \cdot [2\mu_f \in (u_f)] + \nabla p = \rho_f \mathbf{f} \qquad (2.1)$$

and

$$\nabla \cdot u_f = 0 , \qquad (2.2)$$

where $\rho_f$ is the fluid density, $\mu_f$ is the fluid viscosity, $p$ is the pressure, $u_f$ is the fluid viscosity, $u_m$ represents the domain velocity, $\epsilon$ is the velocity strain rate $\epsilon = 1/2(\nabla u_f + \nabla u_f^t)$, $t$ is time and the vector $\mathbf{f}$ represents external forces that may be acting on the fluid, such as gravity or fictitious forces (Coriolis or centrifugal forces).

For the solid mechanics problem [40], the Euler equations (2.3) were discretized using a standard Galerkin method for large deformations with a Newmark time integration scheme. For the deformable solid

$$\rho_s \frac{\partial^2 d_s}{\partial^2 t} = \nabla \cdot P + b, \qquad (2.3)$$

where $\rho_s$ is the solid density, $d_s$ is the displacement field of the solid, $P$ is the first Piola–Kirchhoff stress tensor and $b$ represents the body forces.

The coupling algorithm is a strongly coupled iterative method with Aitken's dynamical relaxation [41]. In one-time step, the fluid mechanics problem is solved, the forces on the coupling surface are calculated and the Aitken's factor is applied. The forces are passed to the solid mechanics code, the body displaces and the new domain location is passed to the fluid mechanics code. This process is repeated until convergence is achieved and then the time step is advanced.

Blood was modelled as an incompressible Newtonian fluid, with constant values of density $1050 \text{ kg m}^{-3}$ and dynamic viscosity $0.035 \text{ kg (m s)}^{-1}$. Blood flow was assumed laminar. The aortic wall was modelled as an isotropic linear solid in FSI simulations. We assumed constant density of $1100 \text{ kg m}^{-3}$, constant patient-specific Young modulus and a Poisson coefficient of 0.45 for the aortic wall. The boundary of the solid domain is divided into inlet, outlets and the FSI interface. The FSI interface is identical to the fluid interface, coupling both domains. The nodes on the interface and inner wall surfaces were defined as traction-free. Conditions of fixed rotation and translation on inlet and outlets cross-sections were imposed in all cases.

Young's modulus is a patient-specific parameter which was calculated iteratively for each patient. The PWV estimated using 4D flow MRI [8] was introduced in the Moens–Korteweg equation [42] (2.4) to obtain an initial Young's modulus $E_0$ which was introduced in the simulations as wall stiffness. After FSI calculations were performed, a new pulse wave velocity (PWV$i$) was calculated by dividing the distance ($\Delta x$) between the sinotubular junction and descending outlet by the wave travel time ($\Delta t$) between systolic peaks at both planes (equation (2.5)). This was repeated (equation (2.6)) until the relative error $\varepsilon$ between PWV of the 4D flow MRI and PWV$_i$ was inferior to $\pm5\%$ (equation (2.7)). Table 2 details the values for PWV extracted from medical images and Young's modulus ($E$) used for each simulation:

$$E_0 = \frac{2 \cdot \text{PWV}_{\text{MR}}^2 \cdot r * \rho}{h}, \qquad (2.4)$$

$$\text{PWV}_i = \frac{\Delta x}{\Delta t_i}, \qquad (2.5)$$

$$E_{i+1} = \frac{2 \cdot \text{PWV}_i^2 \cdot r * \rho}{h} \qquad (2.6)$$

and

$$\varepsilon = \frac{\text{PWV}_i - \text{PWV}_{\text{MR}}}{\text{PWV}_{\text{MR}}}. \qquad (2.7)$$

Time-dependent inlet velocity components measured by 4D flow MRI were interpolated using an in-house code (Matlab, Mathworks, USA) to refine spatial and temporal resolutions from $2.5 \times 2.5 \times 2.5$ mm to $0.25 \times$

**Table 2.** PWV extracted from medical images and Young's modulus ($E$) used for simulation for each studied case.

| | case | PWV (m s$^{-1}$) | $E$ (Pa) |
|---|---|---|---|
| controls | C1 | 7.7 | $1.4 \times 10^6$ |
| | C2 | 11.1 | $2.9 \times 10^6$ |
| | C3 | 6.6 | $9.7 \times 10^5$ |
| | C4 | 6.4 | $8.5 \times 10^5$ |
| stable | S1 | 11.4 | $4.4 \times 10^6$ |
| | S2 | 6.2 | $1.1 \times 10^6$ |
| | S3 | 12.5 | $4.5 \times 10^6$ |
| | S4 | 6.2 | $1.2 \times 10^6$ |
| dilating | D1 | 6.8 | $1.4 \times 10^6$ |
| | D2 | 20.7 | $1.5 \times 10^7$ |
| | D3 | 8.7 | $2.0 \times 10^6$ |
| | D4 | 11.9 | $5.4 \times 10^6$ |

$0.25 \times 0.25$ mm and from 25 to 10 ms, respectively. Figure 1$d$,$f$ presents an example of flow curves at the sinotubular junction and descending aorta.

Patient-specific outlet boundary conditions were applied to ALYA using the 2-element model [23,43], which considers the effect of arterial compliance ($C$), represented as a capacitor and peripheral resistance ($R$) as a resistor:

$$Q(t) = \frac{P(t)}{R} + C\frac{\mathrm{d}P(t)}{\mathrm{d}t}, \tag{2.8}$$

where $Q$ is flow, $P$ is pressure and $t$ is time.

Outlet flow at the descending aorta was measured using MRI and the true flow ratio between the ascending and descending aorta was calculated. $R$ and $C$ were iteratively adjusted on a trial and error basis until the simulated ratio differs less than $\pm 5\%$ from the true ratio.

The analysis of the derived fluid dynamic parameters in the fluid domain was realized with an in-house Matlab code. Flow displacement and jet angle [44] at the distal ascending aorta and proximal descending aorta obtained by CFD and FSI were compared with those measured by 4D flow MRI (figure 1$b$). Flow displacement was calculated by measuring the distance (mm) between the centreline and the location of the maximum velocity of the forward flow at peak systole as described elsewhere [7,44]. Flow angle was calculated by measuring the angle (°) between the centreline and the maximum velocity vector of the forward flow at peak systole [7]. A percentage error is calculated comparing CFD and FSI simulations to 4D flow MRI values. Comparing these parameters can help discerning whether it is critical or not to use the more complex FSI simulations or the simplified CFD simulations. Shear stress ratio (SSR) was analysed at the ascending aorta. SSR is a dimensionless parameter obtained as the ratio of circumferential and axial WSS, thus correlating the in-plane fluid rotation to its through-plane advance:

$$\mathrm{SSR} = \frac{\mathrm{WSS}_{\mathrm{circ}}}{\mathrm{WSS}_{\mathrm{ax}}}, \tag{2.9}$$

where $\mathrm{WSS}_{\mathrm{circ}}$ is the circumferential wall shear stress and $\mathrm{WSS}_{\mathrm{ax}}$ is the axial wall shear stress. SSR evaluates how much the fluid rotates compared to how much it progresses along the vessel, thus compensating for the dependence of axial WSS on the aortic diameter [7,45]. The analysis of SSR consisted of evaluating the statistical differences between FSI and CFD simulations of healthy controls, stable and dilating patients.

Peak systolic SSR was mapped at the wall surface by dividing each circumference in four quadrants using patients' axes: anterior, posterior, left and right as shown in figure 1$c$. Moreover, the ascending aorta surface was divided into proximal and distal to represent the cumulative frequency of both parts.

**Table 3.** Demographics and clinical history. (Mean values are presented with standard deviation and their statistical significance. BSA is body surface area, SBP is systolic blood pressure, DBP is diastolic blood pressure and AAo is the ascending aorta.)

| | | | | p-values | | |
| --- | --- | --- | --- | --- | --- | --- |
| | control | MFS stable | MFS dilating | control versus MFS stable | control versus MFS dilating | MFS stable versus MFS dilating |
| age (years) | 32.5 ± 1.9 | 21.5 ± 1.5 | 35.0 ± 4.5 | 0.019 | 0.384 | 0.081 |
| weight (kg) | 65.0 ± 2.7 | 86.0 ± 5.9 | 70.0 ± 7.3 | 0.029 | 0.773 | 0.083 |
| height (cm) | 176 ± 2 | 185 ± 9 | 179 ± 4 | 0.306 | 0.372 | 0.767 |
| BSA (m$^2$) | 1.78 ± 0.03 | 2.10 ± 0.13 | 1.88 ± 0.10 | 0.083 | 0.773 | 0.110 |
| SBP (mmHg) | 119 ± 3 | 136 ± 6 | 125 ± 9 | 0.083 | 0.773 | 0.386 |
| DBP (mmHg) | 66 ± 6 | 70 ± 4 | 68 ± 5 | 0.773 | 0.772 | 0.767 |
| diameter root (mm) | 29.8 ± 1.4 | 38.4 ± 2.2 | 40.8 ± 2.7 | 0.020 | 0.020 | 0.564 |
| diameter AAo (mm) | 25.5 ± 1.5 | 28.0 ± 1.7 | 36.8 ± 5.8 | 0.144 | 0.108 | 0.146 |

Data are expressed as mean ± s.e.m. Non-parametric Kruskal–Wallis test, followed by a Scheffé's *post hoc* analysis of the original measured values was conducted to determine statistical differences between values. Values of $p < 0.05$ were considered statistically significant.

Original datasets and code are available at https://doi.org/10.5061/dryad.zcrjdfn6j [46].

# 3. Results

CFD and FSI simulations of eight MFS patients and four healthy controls were obtained using MRI-derived patient-specific boundary conditions, geometry and wall stiffness. Flow displacement, jet angle and maximum velocity obtained from CFD and FSI were compared to study the eventual added value of FSI. Afterwards, a WSS-based fluid dynamic marker, SSR, was computed by both CFD and FSI simulations and its capacity to differentiate between MFS patients with and without progressive dilation of ascending aorta aneurysms was tested.

Demographic and clinical data of the subjects are shown in table 3. Overall, we present six men and six women, with a mean age of 32.5 ± 3.8 years in healthy controls, and 21.5 ± 3.0 and 35.0 ± 9.0 years in stable and dilating MFS patients, respectively. Remarkably, demographic and clinical data were similar ($p > 0.05$ in all cases) between stable and dilating patients.

## 3.1. Matching simulations with clinical data

An iterative technique was used to match PWV and descending to ascending aorta flow rates ratio in FSI and CFD simulations to those from MRI.

After three iterations or less, the estimated PWV closely matched the PWV measured *in vivo*. Simulated and measured PWV (figure 2a) were highly correlated ($R^2 = 0.991$). The ratio of descending to ascending aorta flow rates was also compared (figure 2b), obtaining high correlations for both FSI ($R^2 = 0.921$) and CFD ($R^2 = 0.910$)

## 3.2. Computational fluid dynamics versus fluid-structure interaction simulations

Maximum velocity, jet angle and flow displacement from MRI measurements were compared with FSI and rigid wall-CFD simulations results. As shown in table 4, percentage error in flow displacement computation were statistically significantly reduced by more than 40% both in the ascending and the descending aorta comparing FSI with CFD. The jet angle was better estimated with FSI in the ascending and descending aorta, although the reduction with respect to CFD was statistically significant only in the descending aorta. Finally, FSI simulation achieved a maximum velocity percentage error reduction of more than 30% in both planes.

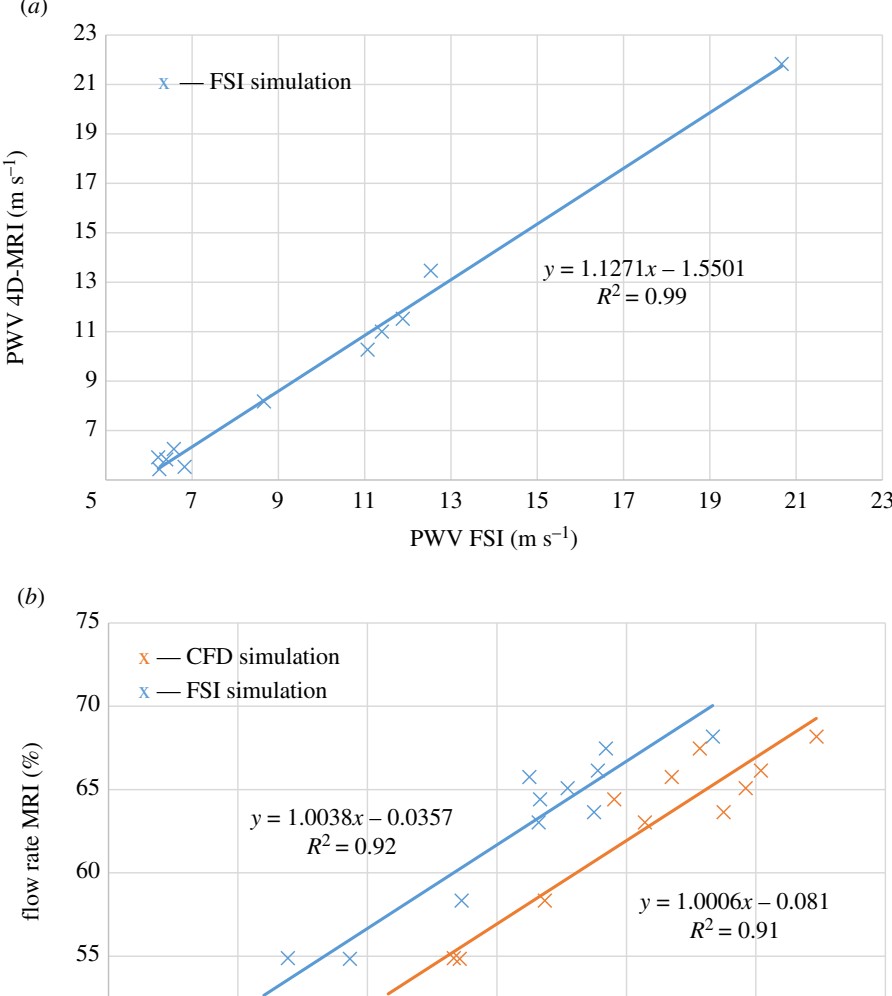

**Figure 2.** (a) Correlation between PWV estimated from 4D flow MRI and using FSI simulations. (b) Correlation between flow rate measured with MRI and flow rate estimated by CFD (orange) and FSI (blue) simulations.

## 3.3. Fluid dynamic markers of aortic dilatation

In this section, results from CFD and FSI simulations are compared with respect to their capacity to stratify aneurysm progression in MFS patients. Considering our prior experience in the association between circumferential WSS and aneurysm as well as the well-known relationship between axial WSS and local diameter [11,35,47], the SSR, a WSS-based marker normalizing circumferential WSS for axial WSS was chosen for the test.

Differences between stable and dilating patients were higher in FSI compared to CFD simulations. FSI-derived SSR being 89% and 81% higher in interior and anterior quarters, respectively, 7% and 8% higher in exterior and posterior quarters, respectively, in dilating compared to non-dilating patients. Differences between dilating and non-dilating MFS patients in FSI-derived SSR were statistically significant in the interior quadrant of ascending aorta. By contrast, CFD-derived SSR was not able to differentiate dilating from non-dilating MFS patients (table 5). Examples of representative distribution of SSR obtained by FSI are reported in figure 3.

The analysis and plotting of the SSR as a cumulative frequency at all ascending aorta nodes in the simulations (figure 4) led to interesting results. In healthy controls, the 90th percentile of the distribution of FSI-derived SSR in the ascending aorta wall was 0.80. In stable patients, the 90th percentile of the distribution of SSR in the ascending aorta wall was 0.80. By contrast, dilating patients

**Table 4.** Flow displacement, jet angle and maximum velocity percentage error with CFD and FSI simulations and percentage error reduction in FSI compared to CFD. (CFD, computational fluid dynamics; FSI, fluid–structure interaction.)

| | CFD | FSI | FSI–CFD (%) | $p$-value |
|---|---|---|---|---|
| ascending aorta percentage error | | | | |
| flow displacement | $18.5 \pm 2.3\%$ | $10.0 \pm 2.5\%$ | 46.0 | <0.01 |
| jet angle | $1.8 \pm 0.5°$ | $1.3 \pm 0.5°$ | 27.7 | 0.18 |
| maximum velocity | $9.1 \pm 1.1\%$ | $5.7 \pm 0.9\%$ | 37.3 | 0.02 |
| descending aorta percentage error | | | | |
| flow displacement | $12.1 \pm 1.2\%$ | $7.1 \pm 1.2\%$ | 40.9 | <0.01 |
| jet angle | $1.3 \pm 0.2°$ | $0.6 \pm 0.1°$ | 50.5 | <0.01 |
| maximum velocity | $11.2 \pm 2.0\%$ | $7.3 \pm 2.1\%$ | 34.6 | <0.01 |

**Table 5.** Systolic peak wall SSR at the ascending aorta in controls, stable patients and dilating patients. ($^{*}p < 0.05$.)

| | | wall shear stress ratio | | | |
|---|---|---|---|---|---|
| | | interior | exterior | anterior | posterior |
| CFD | control | $0.48 \pm 0.16$ | $0.77 \pm 0.22$ | $1.10 \pm 0.46$ | $0.90 \pm 0.30$ |
| | stable | $1.20 \pm 0.75$ | $0.89 \pm 0.36$ | $1.04 \pm 0.27$ | $1.32 \pm 0.20$ |
| | dilating | $1.73 \pm 0.90$ | $1.91 \pm 0.81$ | $1.31 \pm 0.23$ | $5.38 \pm 1.89$ |
| FSI | control | $0.34 \pm 0.10$ | $0.72 \pm 0.25$ | $0.94 \pm 0.52$ | $1.15 \pm 0.28$ |
| | stable | $1.22 \pm 0.73$ | $0.87 \pm 0.34$ | $1.01 \pm 0.20$ | $1.42 \pm 0.18$ |
| | dilating | $2.23 \pm 0.73^{*}$ | $1.97 \pm 0.84$ | $1.50 \pm 0.22$ | $5.83 \pm 1.83$ |

achieved an average SSR cumulative frequency of 90% at 2.38. This is visible in the cases shown in figure 4, where one can observe high shear ratio in the dilating ascending aorta not observed in healthy controls and stable patients. Interestingly, this difference was consistent across controls and patients, as shown in the electronic supplementary material, figures S1–S3. These results indicate that cumulative SSR analysis could be a good tool to stratify aneurysm progression in MFS patients.

# 4. Discussion

Clinical management of aortic aneurysms is driven by aneurysm maximum diameter, but recent studies have reported that the method is not sufficiently accurate. More accurate diagnostic criteria could minimize patients' risk, the amount of unnecessary interventions and healthcare costs, and provide earlier diagnosis of rupture-prone aneurysms. Fluid dynamic parameters could contribute to predict disease progression and to improve rupture risk stratification. Abnormal WSS has been related to aortic wall disruption in bicuspid aortic valve patients [12] or in the presence of aortic stenosis.

In our study, we have demonstrated that: (i) FSI simulations outperform CFD simulations in the computation of jet angle, flow displacement and maximum velocity when compared with 4D flow MRI data in MFS patients, and (ii) the SSR, which is affected by the assumption of rigid wall, especially in FSI simulations, could differentiate between dilating and non-dilating MFS patients. Of note, dilating and non-stable dilating were matched in terms of demographic and clinical condition, highlighting that any aortic root diameter differences were far from being significant.

We have devoted extensive efforts to develop a consistent 4D flow MRI-FSI platform that can replicate aortic flow and wall motion in parallel. 4D flow MRI was used to accurately impose the velocity distribution at the sinotubular junction. The use of Windkessel models led us to match ratio of flow

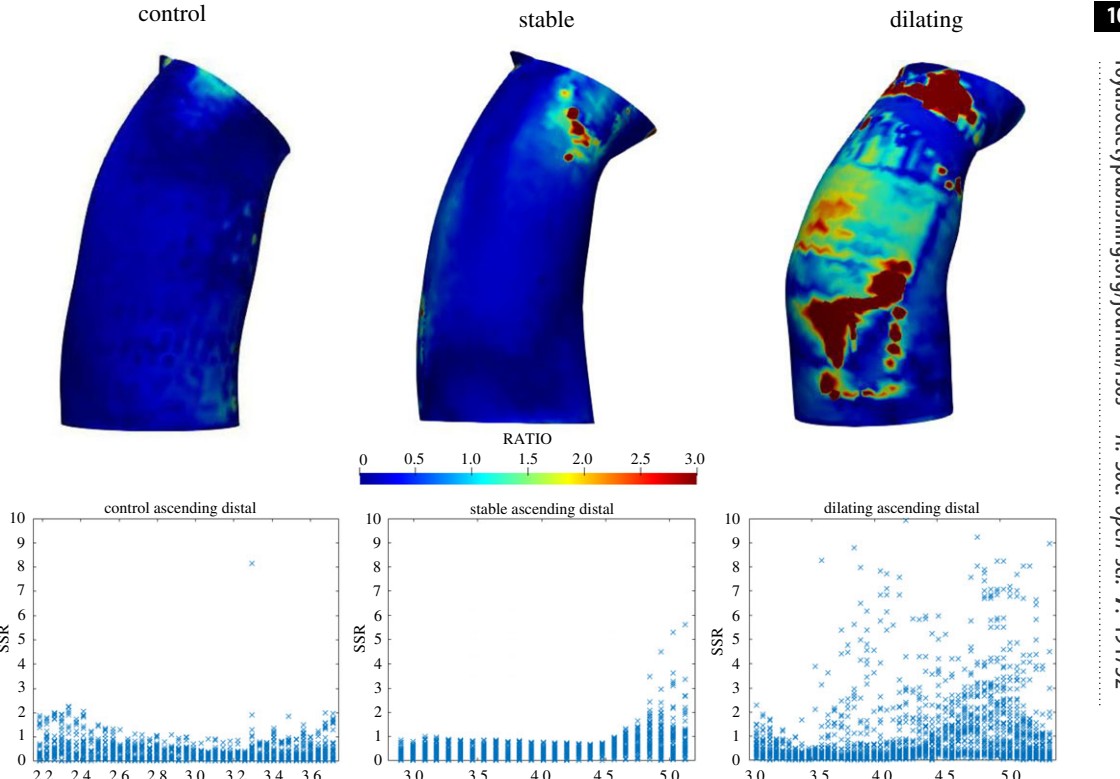

**Figure 3.** Wall SSR maps of the thoracic aorta. From left to right, healthy control number 2, stable patient number 3 and dilating patient number 1. Underneath each simulation, SSR plotted along the distance to the aortic root in the ascending distal segment of the aorta.

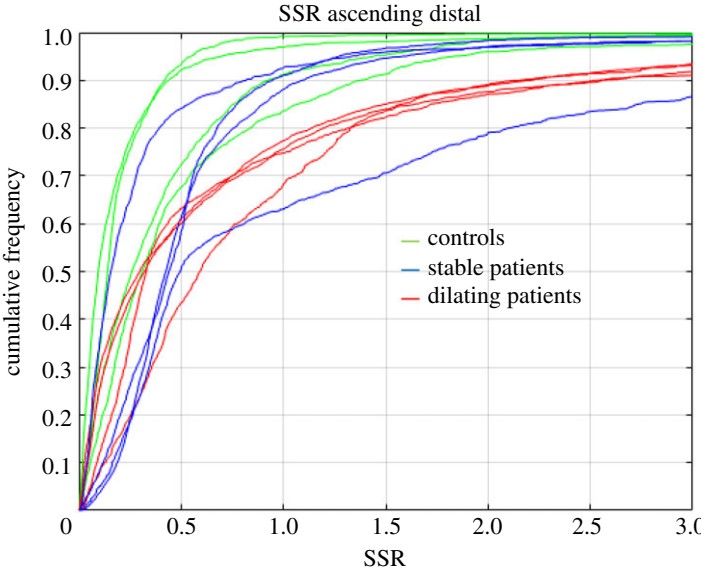

**Figure 4.** Cumulative frequency analysis of SSR at the ascending distal aorta. Healthy controls are drawn in green, stable patients in blue and dilating patients in red.

rates with a quadratic accuracy of more than 90%, guaranteeing that simulated fluid dynamics match closely the 4D flow MRI data. The elasticity of the aorta is another critical parameter to determine its biomechanical status. The use of anisotropic models instead of hyperelastic models, which may better reproduce the movement of the aorta, is a limitation. However, the extraction of patient-specific hypereleastic model parameters is, to the best of our knowledge, not feasible non-invasively. For the first time to our knowledge, we have implemented an iterative technique able to reproduce 4D flow

MRI-derived PWV with 99% accuracy, thus effectively imposing subject-specific, *in vivo* wall elasticity (see the electronic supplementary material). The efficacy of such an approach was remarked by the substantial error reduction in the estimation of aortic haemodynamics. The algorithm used for FSI and the solid and fluid mechanics parts individually, and its implementation in our code ALYA, have already been deeply analysed in previous works [40,48]. There, important matters such as validation, mesh dependency, accuracy, efficiency and implementation aspects, especially those related to the numeric and parallel issues, are widely described.

The presented platform may allow improvements in the estimation of more complex markers by exploiting the best of both *in vivo* imaging techniques and outstanding spatial and temporal resolutions of FSI. Of note, most studies concerning simulation of thoracic aorta flow were performed by CFD alone [15,49,50].

The fluid dynamics markers here compared are those most analysed in 4D flow MRI studies. However, recent studies highlighted the dependence of WSS on spatial and temporal resolution [51]. 4D flow MRI has a spatial resolution of around 2–3 mm voxel$^{-1}$, while spatial resolution of FSI simulation is only limited by computational cost. In our case, we have increased the resolution to 100 μm voxel$^{-1}$ on average, which is simply unachievable using imaging techniques. Increasing the spatial resolution implies a better estimation of flow in the vicinity of the aortic wall, potentially leading to better understanding of aneurysm pathophysiology. By simulating FSI in a computational efficient way, one can overcome spatial and temporal resolutions limitations of 4D flow CMR and deliver a more detailed study. Despite such limitations, 4D flow is the best available technique to non-invasively evaluate velocity field.

In the current study, we observed that SSR values are higher in patients than in controls. Most important, FSI- but not CFD-derived SSR was higher in dilating compared to non-dilating patients. To the authors knowledge, this is the first direct evidence of the impact of rigid wall assumption on the relationship between aneurysm growth and flow characteristics. Although axial and circumferential shear stress have been computed for the SSR calculation, limited differences were observed for both parameters between FSI and CFD simulations and between dilating and non-dilating patients in any directional shear stresses. Of note, the absence of relationship between WSS and progressive dilation found here has already been shown by other authors [52]. The much higher resolution of the FSI and CFD simulations with respect to 4D flow MRI implies that WSS (and derived parameters, such as SSR) cannot be directly compared between *in silico* and *in vivo* studies [51].

The SSR is a dimensionless parameter that normalizes the circumferential WSS with the axial component and reduces the impact of aortic diameter variability. This variability influences axial WSS and can mask pathological conditions. The SSR analysis brings an interesting vision of wall state by showing local shear alterations, i.e. areas where blood does not progress along the vessel but rotates excessively in the plane. We have found that differences in SSR as computed by CFD and FSI simulations are very localized. This observation is only possible if SSR is plotted along the aorta or its cumulative frequency is regionally analysed. Although the focus has been historically on the proximal ascending aorta in MFS patients [22,53], our simulations suggested that the fluid dynamics analysis of the distal ascending aorta may add information, and thus be a powerful tool to improve MFS patients risk stratification.

Our study presents some limitations. Firstly, aortic dilation progression was performed retrospectively, and in a very small number of MFS patients. Results of this study should thus be confirmed by larger, prospective studies. Second, despite the aortic root being the most-commonly affected region in MFS, our analysis was concentrated in the ascending aorta. This was done because of the difficulties to reliably quantify complex flow characteristics in the aortic root by 4D flow MRI, and in the geometrical characterization of this complex region. Moreover, owing to large differences in spatial and temporal resolution between 4D flow and simulations, a comparison of SSR absolute value was not possible [51]. Finally, as aortic wall thickness and stiffness were considered uniform in the aorta, further studies should address the eventual impact of potential thickness and stiffness heterogeneity.

In conclusion, FSI simulations with patient-specific geometry, boundary conditions and aortic wall mechanical properties outperform rigid wall fluid dynamics simulations in the reproduction of thoracic aorta fluid dynamics. SSR, the ratio of circumferential to axial WSS components, depends on the rigid wall assumption and constitutes a good predictor of dilatation in patients with MFS.

Ethics. The study was approved by the Vall d'Hebron Hospital ethics committee named COMITÉ ÉTICO DE INVESTIGACIÓN CLÍNICA CON MEDICAMENTOS del Hospital Universitari Vall d'Hebron. The number code is ID-RTF079. Signed informed consent was obtained from all participants.

Data accessibility. The data of a simulation and code have been added to the Dryad Digital Repository: https://doi.org/10.5061/dryad.zcrjdfn6j [46].

Competing interests. We declare we have no competing interests.

Authors' contributions. R.P. segmented, simulated and post-processed all the data. J.F.R.-P., A.G., G.T.-T., L.D.-S. and A.E. provided the raw data from patients and controls and discussed results. J.C.C., J.J.M. and M.V. provided access to the supercomputing platform and performed fluid dynamic simulations. J.M. oversaw all the work, helped with calculations and validations. R.P., J.F.R.-P., A.G. and J.M. were the major contributors in writing the manuscript. All authors read and approved the final manuscript.

Funding. This study was funded by Ministerio de Economía y Competitividad (grant no. RTC-2016-5152-1), Fundació la Marató de TV3 (grant no. 20151330), FP7 People: Marie-Curie Actions (grant no. 267128), Instituto de Salud Carlos III (grant nos PI14/0106 and PI17/00381) and 'la Caixa' Foundation. M.V. was funded by CompBioMed2, grant agreement ID: 823712, funded under: H2020-EU.1.4.1.3; and SILICOFCM, grant agreement ID: 777204, funded under: H2020-EU.3.1.5.

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
