## [Reviewer comments · Royal Society Open Science]

Review History

RSOS-191752.R0 (Original submission)

Review form: Reviewer 1

Is the manuscript scientifically sound in its present form?

Yes

Are the interpretations and conclusions justified by the results?

Yes

Is the language acceptable?

Yes

Do you have any ethical concerns with this paper?

No

Have you any concerns about statistical analyses in this paper?

No

Reports © 2020 The Reviewers; Decision Letters © 2020 The Reviewers and Editors; Responses © 2020 The Reviewers, Editors and Authors. Published by the Royal Society under the terms of the Creative Commons Attribution License <http://creativecommons.org/licenses/by/4.0/>, which permits unrestricted use, provided the original author and source are credited

Recommendation?

Accept with minor revision (please list in comments)

Comments to the Author(s)

The authors have tried to address all my concerns, the quality of the paper has been improved. I only have some minor comment as bellow,

(1) please explain 'f' in equation 1;

(2) the format of equations, some subscript is in normal font, some in Italic, which is little annoying when reading. It seems the authors used bold face for vector and matrix, but not consistent across the paper, for example equation 3.

(3) Thanks for providing details on obtaining patient-specific Young's modulus from PWV, while I am still not clear how E is optimized, the update of E seems not depending on the difference between PWV_i and PWV_{MR}. In theory, E₀ obtained from equation 5 should give a close value to PWV_{MR}, no further steps are needed. Some explanations will be helpful.

Review form: Reviewer 2

Is the manuscript scientifically sound in its present form?

Yes

Are the interpretations and conclusions justified by the results?

Yes

Is the language acceptable?

Yes

Do you have any ethical concerns with this paper?

No

Have you any concerns about statistical analyses in this paper?

No

Recommendation?

Accept as is

Comments to the Author(s)

The authors have addressed my comments. The manuscript can be accepted by Royal Society Open Science.

Decision letter (RSOS-191752.R0)

20-Dec-2019

Dear Dr Pons

On behalf of the Editors, I am pleased to inform you that your Manuscript RSOS-191752 entitled "Fluid-structure simulations outperform computational fluid dynamics in the differentiation of progressive dilation in Marfan syndrome patients" has been accepted for publication in Royal

Society Open Science subject to minor revision in accordance with the referee suggestions. Please find the referees' comments at the end of this email.

The reviewers and handling editors have recommended publication, but also suggest some minor revisions to your manuscript. Therefore, I invite you to respond to the comments and revise your manuscript.

- Ethics statement

- Data accessibility

If you wish to submit your supporting data or code to Dryad (<http://datadryad.org/>), or modify your current submission to dryad, please use the following link:
<http://datadryad.org/submit?journalID=RSOS&manu=RSOS-191752>

- Competing interests

- Authors' contributions

- Acknowledgements

- Funding statement

Because the schedule for publication is very tight, it is a condition of publication that you submit the revised version of your manuscript before 29-Dec-2019. Please note that the revision deadline will expire at 00.00am on this date. If you do not think you will be able to meet this date please let me know immediately.

Please note that Royal Society Open Science charge article processing charges for all new submissions that are accepted for publication. Charges will also apply to papers transferred to

Royal Society Open Science from other Royal Society Publishing journals, as well as papers submitted as part of our collaboration with the Royal Society of Chemistry (<https://royalsocietypublishing.org/rsos/chemistry>).

If your manuscript is newly submitted and subsequently accepted for publication, you will be asked to pay the article processing charge, unless you request a waiver and this is approved by Royal Society Publishing. You can find out more about the charges at <https://royalsocietypublishing.org/rsos/charges>. Should you have any queries, please contact openscience@royalsociety.org.

on behalf of Dr Francois Fages (Associate Editor) and Marta Kwiatkowska (Subject Editor)
openscience@royalsociety.org

Associate Editor Comments to Author (Dr Francois Fages):

Dear Authors

It is my pleasure to accept your paper with just minor revision for the few points mentioned in the first review.

Thank you for your contribution

Best regards

Reviewer comments to Author:

Reviewer: 1

Comments to the Author(s)

The authors have tried to address all my concerns, the quality of the paper has been improved. I only have some minor comment as bellow,

(1) please explain 'f' in equation 1;

(2) the format of equations, some subscript is in normal font, some in Italic, which is little annoying when reading. It seems the authors used bold face for vector and matrix, but not consistent across the paper, for example equation 3.

(3) Thanks for providing details on obtaining patient-specific Young's modulus from PWV, while I am still not clear how E is optimized, the update of E seems not depending on the difference between PWV_i and PWV_{MR}. In theory, E₀ obtained from equation 5 should give a close value to PWV_{MR}, no further steps are needed. Some explanations will be helpful.

Reviewer: 2

Comments to the Author(s)

The authors have addressed my comments. The manuscript can be accepted by Royal Society Open Science.

Author's Response to Decision Letter for (RSOS-191752.R0)

See Appendix A.

Decision letter (RSOS-191752.R1)

09-Jan-2020

Dear Dr Pons,

It is a pleasure to accept your manuscript entitled "Fluid-structure simulations outperform computational fluid dynamics in the differentiation of progressive dilation in Marfan syndrome patients" in its current form for publication in Royal Society Open Science.

Kind regards,
Lianne Parkhouse
Editorial Coordinator
Royal Society Open Science
openscience@royalsociety.org

on behalf of Dr Francois Fages (Associate Editor) and Professor Marta Kwiatkowska (Subject Editor)
openscience@royalsociety.org

Appendix A

Response to Reviewers

Reviewer: 1

Comments to the Author(s)

The authors have tried to address all my concerns, the quality of the paper has been improved. I only have some minor comment as bellow,

(1) please explain 'f' in equation 1;

We thank the reviewer for their comment and understand the potential confusion. There are two 'f' in equation 1, the subindex (a) and the one at the end of the equation (b).

(a) The subindex 'f' means that the property described is from the fluid. For example, ρ_f is the density of the fluid. This is important as the subindex 'm' is used for the mesh (or domain) velocity (\mathbf{u}_m).

(b) The vector '**f**' represents external forces that may be acting on the fluid, such as gravity or fictitious forces (Coriolis or centrifugal forces).

We have incorporated such explanations in the text, which now reads:

*"where ρ_f is the fluid density, μ_f is the fluid viscosity, p is the pressure, \mathbf{u}_f is the fluid viscosity, \mathbf{u}_m represents the domain velocity, ϵ is the velocity strain rate $\epsilon = \frac{1}{2}(\nabla\mathbf{u}_f + \nabla\mathbf{u}_f^t)$, t is time and the vector **f** represents external forces that may be acting on the fluid, such as gravity or fictitious forces (Coriolis or centrifugal forces)."*

(2) the format of equations, some subscript is in normal font, some in Italic, which is little annoying when reading. It seems the authors used bold face for vector and matrix, but not consistent across the paper, for example equation 3.

We agree with the reviewer. The number of internal iterations in the manuscript has led to some inconsistencies throughout the text. We have corrected the format of the equations and standardized to bold face for vector and matrix. Italic characters have been eliminated from all the equations.

(3) Thanks for providing details on obtaining patient-specific Young's modulus from PWV, while I am still not clear how E is optimized, the update of E seems not depending on the difference between PWV_i and PWV_MR. In theory, E_0 obtained from equation 5 should give a close value to PWV_MR, no further steps are needed. Some explanations will be helpful.

As the reviewer says, in theory, the value for E_0 obtained from equation 5 should give a close value to PWV_{MR} . However, due to the intrinsic (and explained) limitations of the simulations, sometimes the error of PWV_0 was above 5%. Some of these limitations include segmentation model imperfections or using anisotropic model for the solid. The iterative process was used to reduce the difference between PWV from clinical data and simulations

Reviewer: 2

Comments to the Author(s)

The authors have addressed my comments. The manuscript can be accepted by Royal Society Open Science.

We thank the reviewer for their endorsement.

Tracking of changes to the manuscript

- (1) Added the 'f' definition under equation 1.
- (2) Bold face for vector and matrix in equation 3.
- (3) We have changed the text:

"Recently, multiple studies (8,18) have used 4D flow MRI for the assessment of hemodynamics of the aorta evaluating aortic flow patterns, wall shear stress and regional aortic stiffness (via pulse wave velocity) in the thoracic aorta of bicuspid aortic valve patients"

The final form in order to add references (19,20,21) is:

"Recently, multiple studies have used 4D flow MRI for the assessment of hemodynamics of the aorta evaluating aortic flow patterns, wall shear stress and regional aortic stiffness (via pulse wave velocity) in the thoracic aorta of bicuspid aortic valve [8,18–20] and Marfan [21,22] patients."

- (4) Author order list in the web has been matched with the one in the paper.